# A Novel State Space Model with Local Enhancement and State Sharing for Image Fusion

Submission Id: 865

## ABSTRACT

In image fusion tasks, images from different sources possess distinct characteristics. This has driven the development of numerous methods to explore better ways of fusing them while preserving their respective characteristics. Mamba, as a state space model, has emerged in the field of natural language processing. Recently, many studies have attempted to extend Mamba to vision tasks. However, due to the nature of images different from casual language sequences, the limited state capacity of Mamba weakens its ability to model image information. Additionally, the sequence modeling ability of Mamba is only capable of spatial information and cannot effectively capture the rich spectral information in images. Motivated by these challenges, we customize and improve the vision Mamba network designed for the image fusion task. Specifically, we propose the local-enhanced vision Mamba block, dubbed as LEVM. The LEVM block can improve local information perception of the network and simultaneously learn local and global spatial information. Furthermore, we propose the state sharing technique to enhance spatial details and integrate spatial and spectral information. Finally, the overall network is a multi-scale structure based on vision Mamba, called LE-Mamba. Extensive experiments show the proposed methods achieve state-of-the-art results on multispectral pansharpening and multispectral and hyperspectral image fusion datasets, and demonstrate the effectiveness of the proposed approach. Code will be made available.

## CCS CONCEPTS

• **Computing methodologies** → **Reconstruction**.

## KEYWORDS

Multispectral Pansharpening, multispectral and hyperspectral image fusion, Mamba, local-enhanced network

## 1 INTRODUCTION

In the field of image fusion, there are two flourishing tasks that are applied in subsequent applications, *i.e.,* multispectral pansharpening, and multispectral and hyperspectral image fusion. While

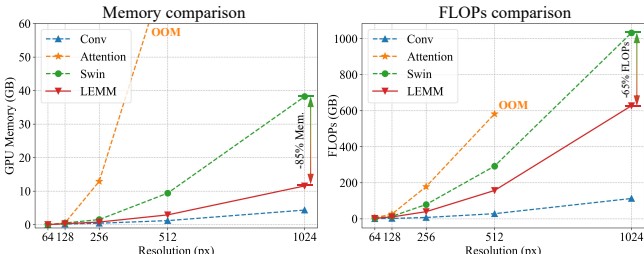

**Figure 1: Memory consumption and FLOPs using different operators on the same U-Net architecture. Our LE-Mamba has linear memory consumption compared with quadratic consumption of self-attention (Attention) [10] and lower FLOPs compared with the attention of Swin Transformer [25].**

multispectral pansharpening aims to fuse high-resolution panchromatic images and low-resolution multispectral images to yield high-resolution multispectral images (HRMS), multispectral and hyperspectral image fusion involves high-resolution RGB images with low-resolution hyperspectral images to yield high-resolution hyperspectral images.

Many deep-learning methods have been widely proposed for image fusion, such as convolution neural networks (CNNs) and vision Transformers. The CNN-based methods can extract and learn local feature information in image fusion. More importantly, vision Transformers have achieved excellent performance in image fusion [5, 38]. However, vision Transformers are limited by the quadratic spatial complexity of the Attention mechanism, making it challenging to process high-resolution images. Many works [14, 25, 40] aim to reduce the complexity of Transformers, but they often come with performance degradation, increased parameter count, and poor generalization capability.

Recently, an alternative approach has shown promising results benefiting from efficiently modeling dependencies of sequence data based on the state space model (SSM). In recent work, [13, 28] introduces the structured state space sequence model as a general sequence model, called Mamba. Afterward, vision Mamba [24, 50] further verifies that it has great potential to be the next-generation backbone for vision foundation models. Compared with vision Transformers [10], vision Mambas adopt sequential visual representation (*i.e.,* image patches) and have subquadratic-time computation and near-linear memory complexity. However, Mamba faces two challenges: ***1)*** *needs for non-unidirectional modeling for 2D images in contrast to language sequence.* ***2)*** *the issue of information stored in the state being lost when the sequence length increases, and* ***3)*** *unable to characterize the spatial and spectral information of images.*

In addressing the first question, two pioneering works stand out: Vim [50], which introduced a bi-directional vision Mamba block

tailored for image processing, and VMamba [24], which proposed a scanning mechanism for images to tackle the causality issue inherent in Mamba's unsuitability for vision tasks. As for the rest of the issues, there has been a lack of exploration of Mamba in the image fusion field. In this paper, we propose two bespoke techniques named local-enhanced vision Mamba (LEVM) block and state sharing to address issues 2) and 3), as its conception shown in Fig. 2, respectively enhancing Mamba's perception of local information and fully exploiting spatial and spectral information in the SSM's state.

The proposed method is more suitable for image fusion tasks with two advantages to the proposed methods. The first advantage is that the proposed LEVM block can represent local and global spatial information. Local information brings rich spatial details, while global information injects long-range pixel dependencies, thus enhancing the detail recovery in the fusion process. The second advantage is that the state sharing method allows information to be shared between layers in the adjacent flow and skip-connected flow. Furthermore, we have also proposed the spatial-spectral learning (S2L) of SSM for the state sharing method. The S2L maps the input image to the state space for spatial representation, and considers the state space as the basis for spectral representation, thereby achieving interaction between spatial and spectral information. To the best of our knowledge, there have been no improvements made to vision Mamba from the perspective of spatial and spectral learning specifically for image fusion tasks. Finally, we propose the LE-Mamba, which is constructed based on the U-Net [33] baseline with the inclusion of the LEVM block and the state sharing technique. The proposed LE-Mamba can achieve superior fusion performance.

The contributions can be summarized as follows:

1) We propose a local-enhanced vision mamba (LEVM) block for the existing vision Mamba architecture to address the issue of spatial information representation.
2) The state sharing technique is designed for the proposed LEVM block, which customizes the image fusion tasks. Then, the state sharing can reduce information loss and enable simultaneous learning of spatial and spectral information within the state space model (SSM).
3) The proposed LE-Mamba achieves state-of-the-art fusion performance for image fusion on four widely-used multispectral pansharpening and hyperspectral multispectral fusion datasets.

The rest of the paper will be organized as follows: In Sect. 2, we review the related work on SSMs, then introduce deep learning-based methods on multispectral pansharpening, as well as multispectral and hyperspectral image fusion tasks. In Sect. 3, we revisit the background knowledge of SSM. Then, in Sect. 4, we elaborate the proposed method with the overall architecture, the local-enhanced vision Mamba block, and the state sharing technique in Sect. 4.1, Sect. 4.2 and Sect. 4.3, respectively. Afterward, Sect. 4.4 analyzes the computational complexity of various common neural network operators. At last, we conduct extensive experiments and ablation studies on the proposed architecture to validate its effectiveness in Sect. 5.

## 2 RELATED WORK

### 2.1 State Space Models

State space models (SSMs) were initially proposed for sequence-to-sequence transformation tasks in natural language processing [12, 13, 28], adept at handling long-range dependencies. In recent years, researchers have strived to address the computational and memory bottlenecks of SSMs in practice, enabling higher efficiency and simplicity when processing long sequences, such as structured state space (S4) [13] introduces parameterization for state space, while Mamba [12] incorporates high-efficient selection mechanism into the S4 based on hardware optimization.

In vision tasks, S4ND method [32] was the first work to introduce SSMs into the vision field, demonstrating their potential to achieve performance on par with models like vision Transformer [10]. VMamba [24] and Vim [50] further applied the ideas of Mamba models to generic vision tasks, proposing mechanisms like bi-directional scanning to better capture image information. Due to the recent explosive growth in research on vision Mamba, we systematically summarize the most recent relevant works in the supplementary for a detailed background.

### 2.2 Deep Learning Methods for Image Fusion

For the multispectral pansharpening task, many CNN-based methods have emerged and obtained promising fused images, including DiCNN [15], PanNet [45], and FusionNet [4]. However, the local feature representation obtained by CNNs hinders better fusion results. For the multispectral and hyperspectral image fusion, there are also many outstanding DL-based works including InvFormer [49] and MiMO-SST [11]. However, these methods bear a significant computational burden due to the quadratic complexity of the Transformer. Despite many efforts to mitigate this issue [14, 25, 40], these methods usually lead to a performance drop.

Recently, SSMs have emerged with near-linear memory consumption and relatively low computational overhead, thus rapidly extending to various vision tasks. Most relative to our work, Pan-Mamba [16] adapted the bidirectional vision Mamba block [50] to the multispectral pansharpening task. *However, it fails to realize the issue of state information loss and does not tailor state space representation to exploit the spatial and spectral domain, leading to suboptimal fusion performance.*

## 3 PRELIMINARY

State space models (SSMs) are a class of sequence models inspired by linear systems, which aim to map a sequence $x(t) \in \mathbb{R}^L$ to $y(t) \in \mathbb{R}^L$ through the hidden space $h'(t), h(t) \in \mathbb{R}^N$. A system matrices $A \in \mathbb{R}^{N \times N}, B \in \mathbb{R}^{N \times 1}$, and $C \in \mathbb{R}^{N \times 1}$ represents the dynamics of the system:

$$h'(t) = Ah(t) + Bx(t),$$
$$y(t) = Ch(t). \tag{1}$$

In practice, the above continuous system should be discretized using zero-order hold assumption, converting the matrics $(A, B)$ to the discretized forms by a timescale $\Delta \in \mathbb{R} > 0$:

$$\overline{A} = \exp(\Delta A)$$
$$\overline{B} = (\Delta A)^{-1}(e^{\Delta A} - I) \cdot \Delta B. \tag{2}$$

The discretized operator $\Delta$ maps $A$ and $B$ from $\Delta : \mathbb{R}^N \to \mathbb{R}^{L \times N}$ (decretize continous system), where dimension $N$ is a higher dimensional latent state. This discretization forms a discretized model written as:

$$h_t = \overline{A} \cdot h_{t-1} + \overline{B} \cdot x_t,$$
$$y_t = C \cdot h_t. \tag{3}$$

Furthermore, in higher-dimensional state space, we can rewrite the above variables, that is input data $x : \mathbb{R}^{L \times D} \to \mathbb{R}^{L \times D \times N}$ with $D$ channel. A timestep $t$ depends on the size of the sequence $L$ (i.e., $x_t \in \mathbb{R}^{D \times N}$). Then, at each timestep $t$, system matrices $\overline{A} \in \mathbb{R}^{D \times N}$, $\overline{B} \in \mathbb{R}^{D \times N}$ are mapped into $N$-dimension state space, and $C \in \mathbb{R}^{D \times N}$ map $h_t$ back the original space. Thus, a hidden state is $h_t \in \mathbb{R}^{D \times N}$ and the output feature is $y_t \in \mathbb{R}^D$ (i.e., $y \in \mathbb{R}^{D \times L}$). According to the implementation of Mamba [12], for efficient computation, the previous computation process can be formulated as the parallel convolution:

$$y = x \circledast \overline{K}$$
$$\text{with} \quad \overline{K} = (C\overline{B}, C\overline{AB}, \cdots, C\overline{A}^{L-1}\overline{B}), \tag{4}$$

where $\circledast$ denotes convolution operation, and $\overline{K}$ is the structured convolutional kernel.

## 4 METHOD

In this section, we will first illustrate the overall network design of LE-Mamba in Sect. 4.1, including the network's input and output, as well as its structure. Subsequently, each local-enhanced vision Mamba (LEVM) block within the network will be delineated in detail in Sect. 4.2. Finally, the implementation details of the proposed state sharing technique are elucidated in Sect. 4.3. In Sect. 4.4, we analyze the complexity of the LEVM block.

## 4.1 Overall Architecture

In some high-level tasks (e.g., classification [24, 50] and segmentation [22, 24, 29, 35]), architectures similar to Meta-formers [46] are commonly used, where plain backbones can offer lower parameter counts, smaller computational loads, and lower latency. However, for the image fusion task, we opted for a multi-scale architecture akin to U-Net [34], which has been proven effective [2]. The designed network is illustrated in Fig. 2(d), featuring an encoder-decoder architecture. The input comprises LRMS and PAN images, while the output is fused images. The encoder and decoder are composed of multiple LEVM layers, with each LEVM layer comprising several LEVM blocks.

For the LEVM block in the encoder, denoted as $f_{enc}(\cdot)$, the input consists of the SSM hidden state $h^{l-1}$ from the previous block, LRMS, and PAN:

$$x^l, h^l = f_{enc}(x^{l-1}, PAN, h^{l-1}), \tag{5}$$

where $l$ represents the $l$-th layer, and $x^0$ denotes LRMS in the first encoder layer (i.e., $l = 0$), then outputs features and hidden states into the next layer, denoted as $x^l$ and $h^l$, respectively. Different from the previous vision Mambas, the state $h^{l-1}$ is incorporated. Its rationale will be explained in Sect. 4.3.

For the LEVM blocks in the decoder, their inputs are the concatenation of the output from the corresponding encoder at the same resolution and the output from the previous decoder layer, denoted as $f_{dec}(\cdot)$. Similar to the LEVM block in the encoder, by incorporating the corresponding encoder's state $h^l_{enc}$, a better fusion performance can be achieved, which is akin to the encoder-decoder skip connections commonly employed in U-Nets. Our intuition is that the encoder's state contains abundant low-level information, which can complement the semantic information in the decoder's state space. This process can be formulated as:

$$x^l, h^l = f_{dec}(concat(x^{l-1}, x^l_{enc}), h^{l-1}, h^l_{enc}). \tag{6}$$

where $x^{l-1}$ and $h^{l-1}$ are input features and hidden states of the $(l-1)$-th layer in the decoder, respectively. Finally, the output of the last LEVM block is mapped back to the pixel space by a linear layer. Following the high-frequency learning strategy proposed in [4], we add the LRMS to the network's output to obtain the fused image. To supervise training, we set the loss function as:

$$\mathcal{L} = \|x^L - GT\|_1 + \lambda \mathcal{L}_{ssim}(x^L, GT), \tag{7}$$

where $GT$ is the ground truth, $L$ denotes the total number of layers of the network, and $\mathcal{L}_{ssim}$ represents loss function based on the SSIM [41]. In practice, we set $\lambda = 0.1$ to balance the two losses.

## 4.2 Local-enhanced Vision Mamba (LEVM)

The current vision Mamba models [24, 50] directly extend the Mamba [12], originally designed for language sequences, to 2D images by patchifying the images into image tokens. According to Eq. (3), each input $x_t$ shares the same system matrices $\overline{A}, \overline{B}, \overline{C}$ to extract global information. This inspires us to study the representation ability of the Mamba model. As shown in Fig. 2(c), we design a local enhancement vision Mamba (LEVM) block to cope with local and global information. The LEVM block has a local and a global VMamba block (i.e., VMambaBlock function, see Fig. 2(b)).

In the local information representation, inspired by Swin Transformer, we partition the image into windows $W \in \mathbb{R}^{p_h \times p_w \times h \times w \times D}$, then feed them into the local VMamba block to extract local information, where $h, w$ denote the window size and there exists $p_h = H/h$, $p_w = W/w$, which can be formulated as:

$$W^{l-1} = WindowPartition(x^{l-1}, (h, w)), \tag{8}$$
$$W^{l-1}, h^l_{local} = VMambaBlock(W^{l-1}, h^{l-1}_{local}). \tag{9}$$

In the global information representation, we can gather the local windows to further represent features from the local VMamba block. Specifically, we utilize the *WindowMerge* function that merges these windows, yielding the entire images, i.e., $x^{l-1} = [W^{l-1}_1, W^{l-1}_2, \cdots, W^{l-1}_{p_h \times p_w}] \in \mathbb{R}^{(p_h \times p_w) \times h \times w \times D}$. For the $l$-th layer, the global VMamba block can be formulated as follows:

$$x^{l-1} = WindowMerge(W^{l-1}, (h, w)) + x^{l-1}, \tag{10}$$
$$x^l, h^l_{global} = VMambaBlock(x^{l-1}, h^{l-1}_{global}). \tag{11}$$

The overall LEVM block can generate local and global state space. The local VMamba block learns local information using SSM and then outputs local windows into the global VMamba block. The global VMamba block combines local information and global information to improve the local representation ability of the original VMamba block.

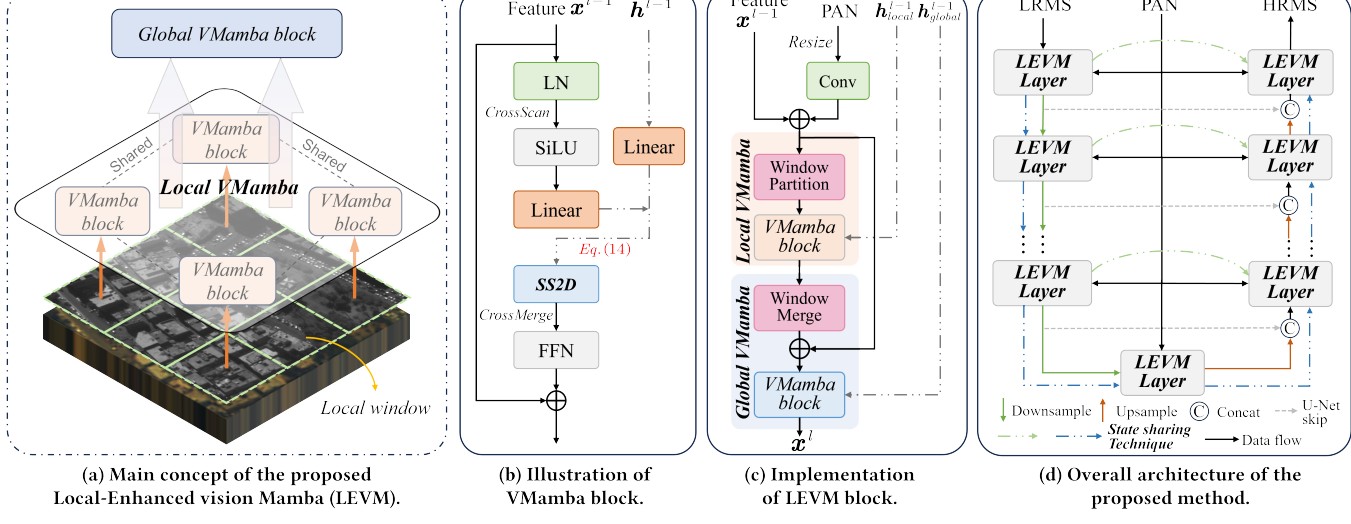

(a) Main concept of the proposed Local-Enhanced vision Mamba (LEVM).

(b) Illustration of VMamba block.

(c) Implementation of LEVM block.

(d) Overall architecture of the proposed method.

**Figure 2: Illustration of (a) the main conception of the proposed LEVM block, (b) the composition of the vision Mamba block, (c) the proposed LEVM block, and (d) the overall architecture of LE-Mamba. "SS2D" in (b) indicates 2D selective scan in [24].**

However, due to the computation limitations of the VMamba block, these SSMs are unable to extract spatial and spectral information separately, resulting in insufficient information fusion. Therefore, we introduce the state sharing technique to achieve interaction between spatial and spectral information. The state sharing technique will be stated in the next section.

### 4.3 State Sharing Technique

In this section, the state sharing technique considers layer-wise information representation and spatial-spectral learning (S2L) of SSM. Specifically, our LE-Mamba uses efficient SSM based on convolution representation to compute image tokens. Then, in a layer of SSM, $\overline{A}$ and $\overline{B}$ are linear transformations, and $C$ sums these states to generate output features. Therefore, the state has a layer-wise information representation. Also, the layer-wise information becomes increasingly semantic from shallow to deep layers.

Inspired by RevCol [1], we aim to retain low-level information across LEVM layers. In contrast to RevCol, which employs deep supervision for each column, we leverage the hidden state of SSM for two information flows propagated across layers, which is a direct and convenient approach. As shown in Fig. 2(d), we build an adjacent flow (denoted as a blue dotted line) and a skip-connected flow (denoted as a green dotted line) to transfer the current state into the next layer and the corresponding decoder layer, respectively. These two information flows are summarized as follows:

(1) The adjacent flow: We can get the local and global hidden states from the LEVM at the $(l-1)$-th layer, i.e., $\boldsymbol{h}_{local}^{l-1}$ and $\boldsymbol{h}_{global}^{l-1}$. Then $\boldsymbol{h}_{local}^{l-1}$ and $\boldsymbol{h}_{global}^{l-1}$ are fed into the next layer.

(2) The skip-connected flow: Instead of the skip-connected method of U-Net, the $(l-1)$-th layer's global hidden state $\boldsymbol{h}_{global}^{l-1}$ is fed into the corresponding decoder layer.

In Fig. 3, we further show feature maps of two information flows containing semantic information and low-level details, respectively.

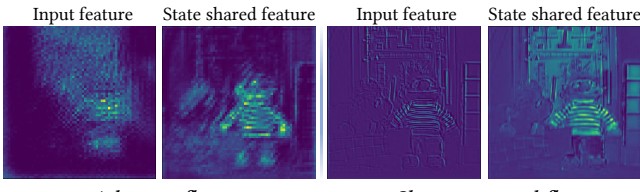

*Adjacent flow*

*Skip-connected flow*

**Figure 3: Illustration of input feature and corresponding state share feature as Eq. (14). Adjacent flow helps to learn more semantic information and skip-connected flow maintains more low-level details and helps fusion. More examples are shown in the supplementary.**

Also, there are some ablation studies (see Tab. 4) to demonstrate the information of adjacent flow and skip-connected can improve fusion performance.

Now, we improve the LEVM block with state sharing. Specifically, we perform a residual connection between the current input features and spatial-spectral learning of SSM (S2L). In S2L, two different linear blocks are conducted to project state space and input features, which can be formulated as follows:

$$\boldsymbol{h}^{l-1} = Linear(\boldsymbol{h}^{l-1}), \quad (12)$$

$$\boldsymbol{g} = Linear(\boldsymbol{x}^{l-1}), \quad (13)$$

where the first linear projects $\boldsymbol{h}^{l-1}$ along the channel dimension, and the second linear projects $\boldsymbol{x}^{l-1} : \mathbb{R}^{D \times L} \rightarrow \mathbb{R}^{N \times L}$.

Although the VMamba block is capable of capturing pixel dependencies well to achieve spatial fidelity, it does not fully exploit spectral information and overlooks the interaction between spatial and spectral information. Based on this, for the $l$-th layer, our S2L decomposes $\boldsymbol{x}^{l-1} \in \mathbb{R}^{D \times L}$ into state space $\boldsymbol{h}^{l-1} \in \mathbb{R}^{D \times N}$ and

**Table 1: Complexity of various vision models. The input image has batch size ($B$), pixel count ($L$), and input channel ($D$) (window-based pixel count $L'$ with $P$ windows), and the hidden dimension is $N$ (that is, the output channel). According to [13], we show parameter counts (Params) and FLOPs, space requirements (Space) for the image fusion task.**

|  | $k \times k$ Conv | Attention | Swin | VMamba | LEVM |
|---|---|---|---|---|---|
| Params | $k^2DN$ | $5DN$ | $5DN$ | $5DN$ | $10DN + D^2 + DN$ |
| FLOPs | $BLk^2DN$ | $B(L^2 + 5LDN)$ | $BP(L'^2 + 5L'DN)$ | $4BLDN + 2LDN$ | $10BLN + 2LD + 2DN$ |
| Space | $BLD$ | $B(L^2 + LD)$ | $BP(L'^2 + L'D)$ | $BLD$ | $BLD$ |

feature space $g \in \mathbb{R}^{N \times L}$. The proposed state sharing can be formulated as:

$$x^{l-1} = x^{l-1} + \alpha h^{l-1} g, \tag{14}$$

where the learnable parameter $\alpha \in \mathbb{R}^D$ is used to balance the information between the S2L and input features. In S2L, the state space is regarded as the basis for spectral representation, and the input image is mapped into the feature space with rich spatial information. Then, we use matrix multiplication to establish the interaction between spectral information and spatial information in the state space. In contrast to the SSM which focuses on processing spatial information, the S2L method takes into account the representation of hidden state in the spectral domain and the spatial characteristics of the image. This allows for extracting and learning two features that simultaneously possess spatial and spectral properties, making it more suitable for image fusion tasks.

Finally, our state sharing technique is summarized in Algorithms 1 and 2. To the best of our knowledge, there haven't been improvements made to the vision Mamba that specifically addresses image fusion tasks. The proposed state sharing technique can be viewed as an independent contribution as it can adapted to any SSMs.

## 4.4 Complexity Analysis

In this section, we analyze the parameter counts and computational complexity of the proposed LEVM with the state sharing method. The parameter count of the local VMamba block and global VMamba block is both $O(5DN)$, respectively. The floating point operations (FLOPs) of these two blocks are $O(5BL'N + 2L'D)$ and $O(5BLN + 2LD)$, respectively. For the state sharing method, the parameter count is $O(D^2 + DN)$ and the FLOPs is $O(DLN)$. The total parameter count is $O(10DN + D^2 + DN)$. The total FLOPs is $O(10BLN + 2LD + 2DN)$. Then, the space requirement is $O(BLD)$. Furthermore, we list these complexity metrics in Tab. 1. In addition, Fig. 1 illustrates that although our LE-Mamba has some increasing parameters and FLOPs than classical methods, we have near-linear memory consumption compared with convolution, self-attention, and attention of Swin Transformer.

## 5 EXPERIMENTS

This section presents extensive experiments on the proposed LE-Mamba, describing the used datasets, and implementation details, and comparing it with previous methods. We analyze the reduced-resolution and full-resolution performances in the multispectral pansharpening task, as well as the performances in the multispectral and hyperspectral image fusion task. Ablation studies are conducted on the proposed LEVM and the state sharing technique to validate their effectiveness.

---

**Algorithm 1:** Mamba parametrization function (`ParamFn`)

---

**Input:** Input feature tokens $x$: `(B, D, L)`, parameters for Param$^A$: `(K, D, N)` and Param$^\Delta$: `(K, D)`.

**Output:** Output parameters $A$: `(K, D, N)`, $B$: `(B, D, N)`, $C$: `(B, D, N)`, $\Delta$: `(B, D, L)`

1 $B \leftarrow \text{Linear}^B(x)$;

2 $C \leftarrow \text{Linear}^C(x)$;

3 $\Delta \leftarrow \log(1 + \exp(\text{Linear}^\Delta(x) + \text{Param}^\Delta))$;

4 $A \leftarrow -\exp(\text{Param}^A)$;

5 **return** $A, B, C, \Delta$

---

**Algorithm 2:** LEVM block with state sharing.

---

**Input:** Last SSM block state $h^{l-1}$: `(B, D, N)`, input feature $x^{l-1}$: `(B, D, L)`.

**Output:** Mamba block output $x^l$, SSM state $h^l$: `(B, D, N)`.

   ▷ Init Param$^A$, Param$^\Delta$ and $D$.

1 Param$^A$: `(K, D, N)`, Param$^\Delta$: `(L, D)` $\leftarrow$ `init`$_{A,\Delta}()$;

2 **def** `forwardFunction`($x^{l-1}, h^{l-1}$):

3     $K \leftarrow 4$;

      ▷ "CrossScan" refers to [24].

4     $x^{l-1}$: `(B, K, D, L)` $\leftarrow$ `CrossScan`($x^{l-1}$);

      ▷ State sharing. "groups" is the group of convolution.

5     $h^{l-1}$: `(B, K, D, N)` $\leftarrow$ `Linear`($h^{l-1}$, groups=K);

6     $g$: `(B, K, N, L)` $\leftarrow$ `Linear`($x^{l-1}$, groups=K);

      ▷ Spatial-spectral learning (S2L) of SSM.

7     $x^{l-1} \leftarrow x^{l-1} +$ `einsum`('BKDN,BKNL→BKDL', $h^{l-1}, g$);

      ▷ Parameterize $A, B, C$ and $\Delta$. See Algo. 1.

8     $A, B, C, \Delta \leftarrow$ `ParamFn`($x^{l-1}$, Param$^A$, Param$^\Delta$);

      ▷ 2D selective scan. Ref to [24].

9     $x^l, h^l \leftarrow$ `SS2D`($x^{l-1}, A, B, C, \Delta$);

      ▷ "CrossMerge" refers to [24].

10    $x^l \leftarrow$ `CrossMerge`($x^l$);

      ▷ Feed forward network with residual connection.

11    $x^l \leftarrow$ `FFN`($x^l$) $+ x^{l-1}$;

12    **return** $x^l, h^l$;

13 **end**

14 **return** `forwardFunction`($x^{l-1}, h^{l-1}$);

---

## 5.1 Dataset and Implementation Details

To evaluate the proposed LE-Mamba, we conduct experiments on two remote sensing multispectral pansharpening datasets: WV3

Table 2: Quantitative results of all competing methods. The best results are in red and the second best results are in blue. Upper panel indicates WV3 (8 bands) dataset and lower panel is GF2 (4 bands) dataset.

| Methods | Reduced-Resolution (RR): Avg±std | | | | Full-Resolution (FR): Avg±std | | | #Params | #FLOPs |
|---|---|---|---|---|---|---|---|---|---|
| | SAM ($\downarrow$) | ERGAS ($\downarrow$) | Q2n ($\uparrow$) | SCC ($\uparrow$) | $D_\lambda$ ($\downarrow$) | $D_s$ ($\downarrow$) | HQNR ($\uparrow$) | | |
| MTF-GLP-FS [39] | 5.32±1.65 | 4.65±1.44 | 0.818±0.101 | 0.898±0.047 | 0.021±0.008 | 0.063±0.028 | 0.918±0.035 | — | — |
| BT-H [26] | 4.90±1.30 | 4.52±1.33 | 0.818±0.102 | 0.924±0.024 | 0.057±0.023 | 0.081±0.037 | 0.867±0.054 | — | — |
| LRTCFPan [43] | 4.74±1.41 | 4.32±1.44 | 0.846±0.091 | 0.927±0.023 | 0.018±0.007 | 0.053±0.026 | 0.931±0.031 | — | — |
| DiCNN [15] | 3.59±0.76 | 2.67±0.66 | 0.900±0.087 | 0.976±0.007 | 0.036±0.011 | 0.046±0.018 | 0.920±0.026 | 0.23M | 0.19G |
| FusionNet [4] | 3.33±0.70 | 2.47±0.64 | 0.904±0.090 | 0.981±0.007 | 0.024±0.009 | 0.036±0.014 | 0.941±0.020 | 0.047M | 0.32G |
| LAGConv [20] | 3.10±0.56 | 2.30±0.61 | 0.910±0.091 | 0.984±0.007 | 0.037±0.015 | 0.042±0.015 | 0.923±0.025 | 0.15M | 0.54G |
| Invformer [48] | 3.25±0.64 | 2.39±0.52 | 0.906±0.084 | 0.983±0.005 | 0.055±0.029 | 0.068±0.031 | 0.882±0.049 | 0.14M | 3.89G |
| DCFNet [42] | 3.03±0.74 | 2.16±0.46 | 0.905±0.088 | 0.986±0.004 | 0.078±0.081 | 0.051±0.034 | 0.877±0.101 | 2.77M | 3.46G |
| HMPNet [37] | 3.06±0.58 | 2.23±0.55 | 0.916±0.087 | 0.986±0.005 | 0.018±0.007 | 0.053±0.006 | 0.929±0.011 | 1.09M | 2.00G |
| PanDiff [31] | 3.30±0.60 | 2.47±0.58 | 0.898±0.088 | 0.980±0.006 | 0.027±0.012 | 0.054±0.026 | 0.920±0.036 | 45.33M | 14.83G |
| PanMamba [16] | 2.94±0.54 | 2.24±0.51 | 0.916±0.090 | 0.985±0.006 | 0.020±0.007 | 0.042±0.014 | 0.939±0.020 | 0.48M | 1.31G |
| Proposed | 2.76±0.52 | 2.02±0.43 | 0.921±0.080 | 0.988±0.003 | 0.016±0.006 | 0.031±0.003 | 0.954±0.007 | 0.74M | 3.58G |
| MTF-GLP-FS [39] | 1.68±0.35 | 1.60±0.35 | 0.891±0.026 | 0.939±0.020 | 0.035±0.014 | 0.143±0.028 | 0.823±0.035 | — | — |
| BT-H [26] | 1.68±0.32 | 1.55±0.36 | 0.909±0.029 | 0.951±0.015 | 0.060±0.025 | 0.131±0.019 | 0.817±0.031 | — | — |
| LRTCFPan [43] | 1.30±0.31 | 1.27±0.34 | 0.935±0.030 | 0.964±0.012 | 0.033±0.027 | 0.090±0.014 | 0.881±0.023 | — | — |
| DiCNN [15] | 1.05±0.23 | 1.08±0.25 | 0.959±0.010 | 0.977±0.006 | 0.041±0.012 | 0.099±0.013 | 0.864±0.017 | 0.23M | 0.19G |
| FusionNet [4] | 0.97±0.21 | 0.99±0.22 | 0.964±0.009 | 0.981±0.005 | 0.040±0.013 | 0.101±0.013 | 0.863±0.018 | 0.047M | 0.32G |
| LAGConv [20] | 0.78±0.15 | 0.69±0.11 | 0.980±0.009 | 0.991±0.002 | 0.032±0.013 | 0.079±0.014 | 0.891±0.020 | 0.15M | 0.54G |
| Invformer [48] | 0.83±0.14 | 0.70±0.11 | 0.977±0.012 | 0.980±0.002 | 0.059±0.026 | 0.110±0.015 | 0.838±0.024 | 2.77M | 3.46G |
| DCFNet [42] | 0.89±0.16 | 0.81±0.14 | 0.973±0.010 | 0.985±0.002 | 0.023±0.012 | 0.066±0.010 | 0.912±0.012 | 2.77M | 3.46G |
| HMPNet [37] | 0.80±0.14 | 0.56±0.10 | 0.981±0.030 | 0.993±0.003 | 0.080±0.050 | 0.115±0.012 | 0.815±0.049 | 1.09M | 2.00G |
| PanDiff [31] | 0.89±0.12 | 0.75±0.10 | 0.979±0.010 | 0.989±0.002 | 0.027±0.020 | 0.073±0.010 | 0.903±0.021 | 45.33M | 14.83G |
| PanMamba [16] | 0.68±0.12 | 0.64±0.10 | 0.982±0.008 | 0.985±0.006 | 0.016±0.008 | 0.045±0.009 | 0.939±0.010 | 0.48M | 1.31G |
| Proposed | 0.60±0.11 | 0.52±0.09 | 0.987±0.007 | 0.994±0.001 | 0.018±0.009 | 0.027±0.008 | 0.955±0.011 | 0.74M | 3.58G |

and GF2 datasets with super-resolved scale 4, and further on indoor hyperspectral-multispectral fusion datasets: CAVE and Harvard datasets with scale 8. More details on the datasets can be found in the supplementary.

During the training process, we use the AdamW [27] optimizer and set the base learning rate to 1e-3, which is decreased to 1e-4 at 300 epochs, and further decreased to 1e-5 at 600 epochs. After that, the training continues till 1000 epochs for WV3 and GF2 datasets and 1600 epochs for CAVE and Harvard datasets. The weight decay is set to 1e-6. All experiments are conducted on two RTX 3090 GPUs. Network configurations on different datasets can be found in the supplementary.

## 5.2 Benchmark

For the multispectral pansharpening datasets, we compare the proposed method with the recently SOTA traditional methods: MTF-GLP-FS [39], BT-H [26] and LRTCFPan [43], and DL-based methods: DiCNN [15], LAGConv [20], DCFNet [42], HMPNet [37], PanDiff [31] and PanMamba [16]. For the multispectral and hyperspectral datasets, we also choose some recent traditional methods including CSTF-FUS [21], LTTR [8], LTMR [7] and IR-TenSR [44]. Moreover, most competitive DL-based methods are compared: ResTFNet [23], SSRNet [47], HSRNet [18], MogDCN [9], Fusformer [17], DHIF [19], PSRT [6], 3DT-Net [30], DSPNet [36], and MIMO-SST [11].

## 5.3 Main Results

The fusion performance of LE-Mamba on the WV3 and GF2 datasets is provided in Tab. 2, indicating the SOTA performance of our method in reduced-resolution metrics. Traditional approaches lag significantly behind DL-based methods in terms of reduced-resolution metrics. Among DL-based methods, our approach demonstrates good fidelity for both spectral modality (*i.e.,* SAM metric) and spatial modality (*i.e.,* ERGAS metric). In terms of full-resolution metrics, LE-Mamba exhibits SOTA performance on the GF2 test set, indicating its good generalization capability. We present fusion images on the WV3 test set in Fig. 4, showcasing LE-Mamba's minimal errors, particularly evident in the fusion of high-frequency components such as the edges of buildings and roads.

Tab. 3 provides widely used numerical fusion metrics on CAVE and Harvard datasets. Our LE-Mamba outperforms all previous traditional and DL-based methods contributed by the multi-scale backbone and proposed LEVM equipped with state sharing technique. Error maps are depicted in Fig. 5 which shows that LE-Mamba owns fewer errors and better fusion performances. More visual results can be found in the supplementary.

## 5.4 Ablation Study

*5.4.1 LEVM and State Sharing Technique can Boost Fusion Performance.* We conduct ablation studies on the proposed local-enhanced

**Table 3: The average and standard deviation calculated for all the compared approaches on 11 CAVE examples and 10 Harvard examples simulating a scaling factor of 8. The best results are in red and the second best results are in blue.**

| Methods | CAVE ×8 | | | | Harvard ×8 | | | | #Params | #FLOPs |
|---|---|---|---|---|---|---|---|---|---|---|
| | PSNR (↑) | SAM (↓) | ERGAS (↓) | SSIM (↑) | PSNR (↑) | SAM (↓) | ERGAS (↓) | SSIM (↑) | | |
| Bicubic | 29.96±3.54 | 5.89±2.32 | 5.56±3.08 | 0.887±0.066 | 33.18±6.85 | 3.10±0.90 | 3.83±1.84 | 0.894±0.0732 | – | – |
| CSTF-FUS [21] | 38.44±4.25 | 7.00±2.65 | 2.11±1.15 | 0.959±0.033 | 39.84±6.51 | 4.49±1.52 | 2.40±1.84 | 0.932±0.092 | – | – |
| LTTR [8] | 37.92±3.59 | 5.37±1.96 | 2.44±1.05 | 0.972±0.018 | 42.09±4.56 | 3.62±1.34 | 1.80±0.96 | 0.960±0.048 | – | – |
| LTMR [7] | 38.41±3.57 | 5.04±1.70 | 2.24±0.97 | 0.974±0.017 | 42.09±4.56 | 3.62±1.34 | 1.80±0.92 | 0.959±0.060 | – | – |
| IR-TenSR [44] | 36.79±3.64 | 12.87±4.98 | 2.68±1.41 | 0.944±0.031 | 40.04±3.89 | 5.40±1.76 | 4.75±1.55 | 0.958±0.016 | – | – |
| ResTFNet [23] | 43.77±5.34 | 3.49±0.94 | 1.38±1.25 | 0.992±0.006 | 43.50±3.96 | 3.53±1.11 | 1.74±0.93 | 0.979±0.012 | 2.387M | 1.75G |
| SSRNet [47] | 46.23±4.19 | 3.13±0.97 | 1.05±0.73 | 0.993±0.004 | 45.76±3.34 | 2.99±0.98 | 1.34±0.74 | 0.983±0.010 | 0.027M | 0.11G |
| HSRNet [18] | 46.69±4.48 | 2.91±0.86 | 0.93±0.63 | 0.994±0.003 | 44.02±4.89 | 3.64±1.79 | 1.49±0.81 | 0.980±0.013 | 1.09M | 2.00G |
| MogDCN [9] | 49.21±4.99 | 2.44±0.74 | 0.76±0.63 | **0.996±0.003** | 45.14±5.41 | 3.19±1.45 | 1.75±1.66 | 0.980±0.019 | 6.840M | 47.48G |
| Fusformer [17] | 47.96±7.79 | 2.75±1.30 | 1.45±2.69 | 0.990±0.022 | 44.93±5.65 | 3.63±2.40 | 1.49±0.96 | 0.979±0.017 | 0.504M | 9.83G |
| DHIF [19] | 48.46±4.89 | 2.50±0.79 | 0.83±0.67 | **0.996±0.003** | 45.00±4.13 | 3.70±1.68 | **1.32±0.61** | 0.983±0.011 | 22.462M | 54.27G |
| PSRT [6] | 47.86±7.53 | 2.73±0.80 | 1.52±3.02 | 0.994±0.005 | 45.10±4.06 | **2.90±0.84** | 1.37±0.84 | **0.985±0.009** | 0.247M | 1.14G |
| 3DT-Net [30] | **49.41±5.83** | **2.26±0.66** | 0.83±1.07 | **0.996±0.003** | 44.41±5.38 | 2.93±0.88 | 1.55±0.89 | 0.983±0.010 | 3.464M | 68.07G |
| DSPNet [36] | 49.18±4.84 | 2.57±0.79 | **0.75±0.62** | **0.996±0.003** | 45.84±3.62 | 2.97±0.75 | 1.33±0.64 | 0.984±0.010 | 6.064M | 6.81G |
| MIMO-SST [11] | 48.31±5.04 | 2.88±0.86 | 0.89±0.79 | 0.995±0.004 | **46.59±3.34** | 2.91±0.75 | 2.29±1.03 | **0.985±0.009** | 4.983M | 1.58G |
| Proposed | **49.86±4.77** | **2.31±0.69** | **0.70±0.56** | **0.997±0.002** | **46.84±3.82** | **2.75±0.71** | **1.16±0.53** | **0.986±0.009** | 3.158M | 9.80G |

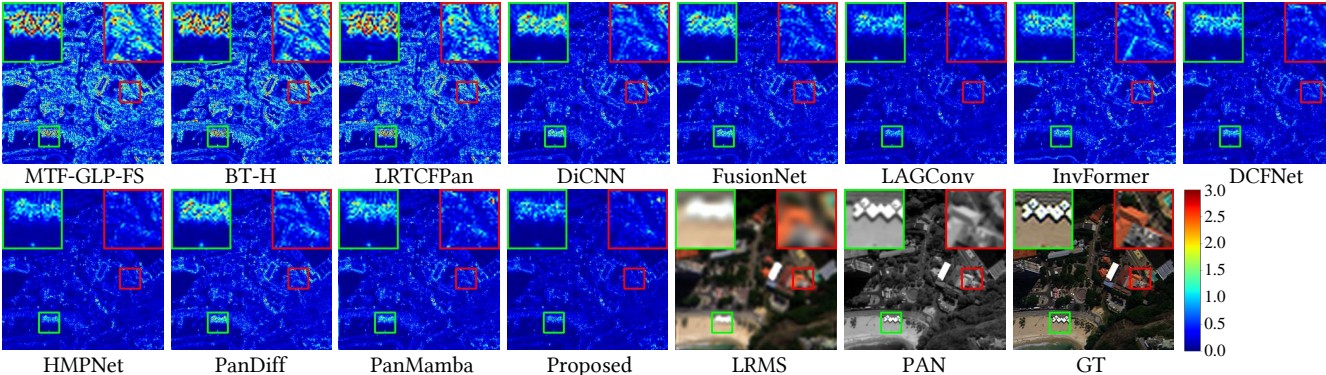

MTF-GLP-FS · BT-H · LRTCFPan · DiCNN · FusionNet · LAGConv · InvFormer · DCFNet

HMPNet · PanDiff · PanMamba · Proposed · LRMS · PAN · GT

**Figure 4: Error maps of LE-Mamba and compared previous SOTA methods on WV3 test set. Our LE-Mamba shows fewer errors on GT. Some close-ups are depicted at the corner.**

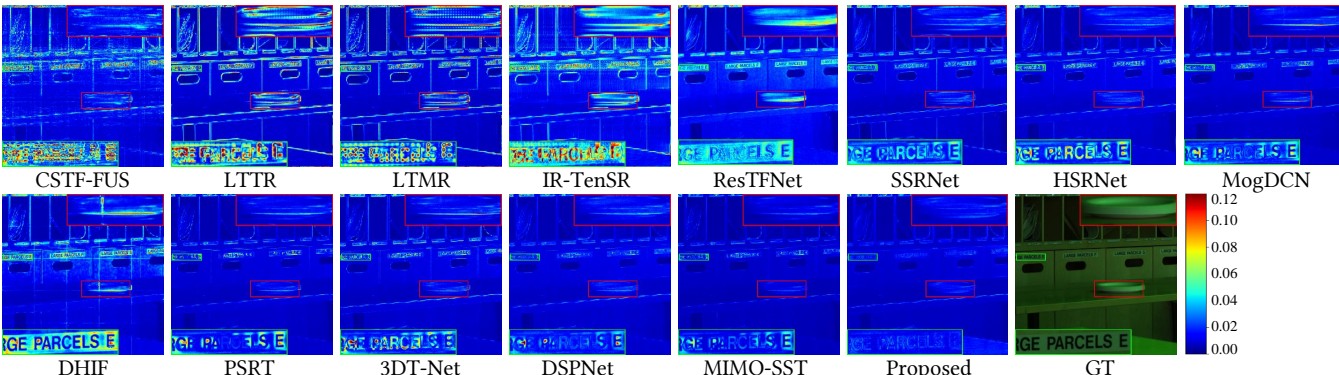

CSTF-FUS · LTTR · LTMR · IR-TenSR · ResTFNet · SSRNet · HSRNet · MogDCN

DHIF · PSRT · 3DT-Net · DSPNet · MIMO-SST · Proposed · GT

**Figure 5: Error maps of LE-Mamba and compared previous SOTA methods on Harvard (× 8) test set. Our LE-Mamba shows fewer errors on GT. Some close-ups are depicted at the corner.**

vision Mamba block and state sharing technique and report the fusion performance on Tab. 4. Starting at the basic multi-scale network baseline which only contains NAFBlock [3], we observe the performance is poor whose SAM only has 3.52. Then we replace the NAFBlock with the LEVM block, the performance is boosted to 2.86. Based on it, we further share the SSM state at adjacent flow and skip-connected flow and conduct spatial-spectral learning (S2L), one can notice the fusion performance is better and the SAM metric reaches 2.76. From the consistent performance improvement in the above experiments, we can summarize that our proposed LEVM and state sharing technique are effective in boosting the performance of the image fusion task. We will conduct ablation studies on the S2L method in the next round.

**Table 4: Ablation study on proposed multi-scale architecture, LEVM block, and state sharing technique (including adjacent flow and skip-connected flow) on WV3 dataset.**

| Composition | SAM ($\downarrow$) | ERGAS ($\downarrow$) | Q2n ($\uparrow$) | SCC ($\uparrow$) |
|---|---|---|---|---|
| Multi-scale baseline | 3.52 | 2.66 | 0.902 | 0.977 |
| + VMamba | 2.93 | 2.15 | 0.910 | 0.987 |
| + LEVM block | 2.86 | 2.17 | 0.914 | 0.986 |
| + adjacent flow | 2.80 | 2.08 | 0.918 | 0.987 |
| + skip-connected flow | 2.76 | 2.02 | 0.921 | 0.988 |

*5.4.2 State Sharing Beats Non-state Residual Connection.* From Eq. (14), we can observe that the state sharing technique is a *residual connection* that adds input feature $x^{l-1}$ with S2L (*i.e.,* $h^{l-1}g$). To verify the effectiveness of using the S2L (whether the performance gain comes from residual connections and additional convolutional parameters), we design another residual variant that *does not use the SSM state*:

$$x^{l-1} = x^{l-1} + \alpha Linear(x^{l-1}). \quad (15)$$

We validate its fusion performance on the WV3 dataset, as shown in Tab. 5. Our state sharing outperforms the residual variant. From this ablation, we can conclude that: using S2L to share state across different blocks can bring performance gain.

**Table 5: Discussion on the S2L of the proposed LE-Mamba on the WV3 dataset.**

| Variants | SAM ($\downarrow$) | ERGAS ($\downarrow$) | Q2n ($\uparrow$) | SCC ($\uparrow$) |
|---|---|---|---|---|
| $x^{l-1}$ | 2.86 | 2.17 | 0.914 | 0.986 |
| $x^{l-1} = x^{l-1} + \alpha Linear(x^{l-1})$ | 2.84 | 2.13 | 0.917 | 0.986 |
| $x^{l-1} = x^{l-1} + \alpha h^{l-1}g$ | 2.76 | 2.02 | 0.921 | 0.988 |

*5.4.3 LEVM Outperforms Previous Attentions.* To compare the proposed LEVM with previously common Attention mechanisms, we select common attention and its variants including traditional Attention [10], PVT Attention [40], Swin Attention [25], and Linear Attention [14]. We replace the LEVM block with the corresponding Attention block (including FFN) and train them uniformly on the WV3 dataset until convergence. Their fusion performance is shown in Tab. 6. It can be seen that the traditional Attention performs the worst, which is because using Attention in the first layer

of the network occupies an extremely large GPU memory, leading to the inability to train. Therefore, most methods based on traditional Attention perform patch embedding in the first layer to downsample the image size, which is disastrous for tasks like image fusion that require low-level information. The performance of Linear Attention is also relatively poor, which is caused by the approximation errors introduced by its approximation of the Attention mechanism. Next are PVT Attention and Swin Attention, both of which impose restrictions on Attention. PVT downsamples both the keys and values, resulting in information loss, while Swin limits the global information to the local regions. The proposed LEVM outperforms previous attention operations.

**Table 6: Results of *different self-attention (attention) types* on WV3 dataset.**

| Attention types | SAM ($\downarrow$) | ERGAS ($\downarrow$) | Q2n($\uparrow$) | SCC ($\uparrow$) |
|---|---|---|---|---|
| Self-attention [10] | 3.09 | 2.33 | 0.906 | 0.917 |
| PVT attention [40] | 2.89 | 2.16 | 0.919 | 0.986 |
| Swin attention [25] | 2.86 | 2.15 | 0.919 | 0.986 |
| Linear attention [14] | 2.94 | 2.21 | 0.914 | 0.984 |
| Proposed | 2.76 | 2.02 | 0.921 | 0.988 |

*5.4.4 Enlarging SSM State Capacity Does not Solve Issue 2).* To solve issue 2) mentioned in Sect. 1, one simple and straightforward idea is to enlarge the SSM state capacity $N$. Usually, $N$ is set to 16 and suits most high-level tasks. We design a series of enlarged SSM state variants with $N = 32$, $N = 64$ based on Tab. 4 "+VMamba" model. One can observe that even when setting $N = 64$ with a large state capacity, the performance gain is limited. Moreover, large $N$ brings additional memory consumption and FLOPs. It is unacceptable for resource-constraint scenarios.

**Table 7: The performance of LE-Mamba with different SSM states capacity on WV3 datasets.**

| Variants | SAM ($\downarrow$) | ERGAS ($\downarrow$) | Q2n ($\uparrow$) | SCC ($\uparrow$) |
|---|---|---|---|---|
| $N = 16$ | 2.93 | 2.15 | 0.910 | 0.987 |
| $N = 32$ | 2.90 | 2.14 | 0.914 | 0.987 |
| $N = 64$ | 2.88 | 2.13 | 0.915 | 0.987 |

## 6 CONCLUSION

Based on the observation that sequence modeling in the Mamba network doesn't fully consider the characteristics of the image fusion task, this paper designs a multi-scale network architecture equipped with the local-enhanced vision Mamba (LEVM) block and the state sharing technique, called LE-Mamba. The LEVM mitigates the information loss caused by limited states in the Mamba network by combining local and global information to enhance fused image details. Additionally, the state sharing technique shares the state between layers in adjacent and skip-connected flows, enabling the deep layers of the network to contain richer spatial information. Then, the fused image exhibits improved spatial and spectral details by incorporating spatial-spectral learning (S2L). The proposed network architecture achieves state-of-the-art performance in widely used image fusion datasets.

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
