# OpenReview forum: "A Novel State Space Model with Local Enhancement and State Sharing for Image Fusion"
_acmmm.org/ACMMM/2024/Conference — MM2024 Poster_

### Official Review · Reviewer_ZfhQ · 2024-05-16

**Rating:** 5
**Confidence:** 4

**Summary:**

This paper proposes a local-enhanced Mamba network and an accompanying state sharing technique that achieves state-of-the-art results on remote sensing pansharpening and hyperspectral image fusion tasks. The ablation study also shows that incorporating the local-enhanced Mamba block performs better than other attention or convolutional operations. The paper also ablates the proposed state sharing technique, demonstrating its effectiveness.

**Strengths:**

1. It's great that the paper has clear motivation, good writing, and positive illustrations explaining the role of each module.
2. The superior performance of the proposed LE-Mamba over other methods is a compelling result. Besides, the state-sharing technique proposed is novel and ingenious in addressing the limited state capacity issue faced by SSM-based networks for low-level tasks. The experimental validation of its effectiveness is a good point.
3. It is very commendable that authors compare different attention operations within the same network architecture, and analyse their tradeoffs in terms of memory consumption and FLOPs. The given systematic evaluation is valuable for understanding the pros and cons of different choices.

**Limitations:**

1. PanDiff has such a large number of parameters, why does it perform poorly, and what is different between it and other methods?
2. What does "Space" in Tab.1 refer to? It should be memory consumption?
3. In Alg. 2 Line 5, why does Linear have groups, perhaps it is grouped convolution, this is not a technical issue, but the authors need to state it precisely.
4. In Sect. 2.2 Line 208, it writes "near-linear", but the caption in Fig.1 writes "linear". I think it should be corrected to also "near-linear" in the caption according to the plot.

**Suitability:**

3

---

### Official Review · Reviewer_WedV · 2024-05-16

**Rating:** 5
**Confidence:** 4

**Summary:**

This paper achieves new SOTA performance on image fusion task by lifting Mamba model with a new local-enhancement Mamba module and a state sharing technique. By comparing with previous commonly-used attention and convolution operators, the proposed LEVM shows promising multi-modal fusion performance and memory, FLOPs balance. Some illustrations of state sharing technique shows convinced motivation. Overall, I think this is a novel paper that improves vision mamba and helps multi-modal image fusion.

**Strengths:**

1. This paper notes two issues that vision mamba models face. And the authors proposed two helpful improvements to mitigate these issues. The results and module ablation studies prove it.
2. Local-enhancement mamba module mitigate the information loss.
3. Clear illustrations of adjacent flow and skip-connected flow shows the effectiveness on feature map.
4. Relatively speaking, the proposed LEVM achieves a good balance between convolution and attention.

**Limitations:**

1. The authors should make implementation of FFN clear, since it is also a prominent component of the proposed LEVM.
2. Why 3DT-Net owns 68.07G FLOPs, far more than other methods? Is this a typo? Correct me if I am wrong.
3. Since this paper is about SSM, the hyperspectral and multispectral image fusion benchmark lacks a SSM counterpart, the author should consider to include a SSM architecture to compare.

typos: (1) In Figure 2 (b), should the caption be "... of LEVM block"? (2) In Table 1, the "Space" indicates "GPU Memory"?

**Suitability:**

3

---

### Official Review · Reviewer_jw1W · 2024-05-16

**Rating:** 5
**Confidence:** 4

**Summary:**

This paper proposed a novel state space model with local-enhancement and state-sharing technique to enhance the performance of the vision Mamba on image fusion tasks. The proposed local-enhanced vision Mamba alleviates the state information loss issue of SSMs mentioned in the paper. Additionally, the proposed state-sharing technique can be applied to other SSMs to boost their performance. The conducted experiments on image fusion tasks, and the promising results demonstrate the effectiveness of their proposed methods.

**Strengths:**

(1) The proposed method reaches a new SOTA in image fusion task.
(2) This paper starts from the issue of insufficient SSM state capacity, and considers enhancing it to local-enhanced Mamba: assigning each window to a local Mamba for processing, and finally processing global information with another global Mamba block, which is innovative.
(3) The authors propose a state sharing technique, sharing the local and global states of each LEVM between adjacent and skipped blocks, solving the issue 3) considered in the paper.

**Limitations:**

(1) Akin to Swin transformer, the author should consider to shift the window to make the information communication in each window (although there still exists the global mamba block).
(2) The author should explain how to visualize the proposed adjacent flow and skip-connected flow.
(3) What does the $K\leftarrow 4$ in Algo.2 Line 3 means?
typos: 1) "casual" in abstract, should it be "causal"?

**Suitability:**

3

---

### Official Review · Reviewer_Tif7 · 2024-05-23

**Rating:** 3
**Confidence:** 3

**Summary:**

Considering the characteristics of the image fusion task, this paper proposes a multi-scale network with local enhancement and state sharing. The network comprehensively preserves the details of the source image by extracting global and local information. State-sharing technology shares the state between adjacent and skip-connected flows, reducing information loss and learning spatial and spectral information simultaneously.

**Strengths:**

1. Corresponding improvements have been made to address the shortcomings of Mamba in image fusion. Mamba blocks inevitably suffer information loss as the sequence length increases. The authors solve this problem by developing state-sharing technology, which applies the hidden state of SSM for two information flows propagated across layers. The authors constructed adjacent flow and skip connection flow transfer states to achieve competitive performance.
2. Extensive experiments have proven the superiority of LE-Mamba. From the qualitative and quantitative experimental results given in the paper, the performance of LE-Mamba has reached SOTA. In addition, the author conducted ablation experimental studies for his innovative design, and the results clearly demonstrate that the proposed technique is effective.

**Limitations:**

1. For local-enhanced vision Mamba (LEVM), the authors are inspired by the Swin Transformer and partition the image into windows, then feed them into the network. So if the author has made corresponding improvements to the LEVM for image fusion tasks, is the LEVM just a migration application of Swin Transformer?

2. The author said in the introduction that one of the drawbacks of Mamba is that it is unable to characterize spatial and spectral information. Please explain what flaw in Mamba causes this situation. How does the S2L learning developed by the author achieve the characterization?

3. I wonder what the advantages of applying the state space model to image fusion are compared to general CNN.

4. The obvious advantage of Mamba is its efficiency. As can be seen from the authors' quantitative experiments, the efficiency of LE-Mamba is competitive. However, FLOPS is related to many factors and cannot fully prove the efficiency of the network. It is recommended to add time as a supplement to demonstrate the advantages of LE-Mamba.

5. When describing the advantages of his method, the author mentioned that the interaction of spatial and spectral information can be achieved. As can be seen from Figure 2(c), the author simply adds the two features and sends them into VMamba. Is this how the author implements interaction? If not, please add a detailed explanation in the paper.

**Suitability:**

3

---

### Meta-Review · Area_Chair_ZcuY · 2024-06-30

**Recommendation:** Accept (Poster)
**Confidence:** 5

**Metareview:**

This submission has been reviewed by four experts, the final ratings from whom are all positive after rebuttal (although one of them held a negative vote in initial phase, after rebuttal, the reviewer raised his/her score). Considering the quality of the submission, the comments from the reviewers, and the rebuttal from the authors, the paper can be accepted by the conference.